# Biogeographic pattern of microbial communities inhabiting terrestrial mud volcanoes across the Eurasian continent

Tzu-Hsuan Tu[1,2,3], Li-Ling Chen[2], Yi-Ping Chiu[3], Li-Hung Lin[3,4], Li-Wei Wu[5], Francesco Italiano[6], J. Bruce H. Shyu[3], Seyed Naser Raisossadat[7,8], and Pei-Ling Wang[2,4]*

[1]Department of Oceanography, National Sun Yat-sen University, Kaohsiung, Taiwan
[2]Institute of Oceanography, National Taiwan University, Taipei, Taiwan
[3]Department of Geosciences, National Taiwan University, Taipei, Taiwan
[4]Research Center for Future Earth, National Taiwan University, Taipei, Taiwan
[5]Department of Life Science, Tunghai University, Taichung, Taiwan
[6]National Institute of Geophysics and Volcanology, Palermo, Italy
[7]Department of Geology, University of Birjand, Birjand, Iran
[8]Earth Science Research Group, University of Birjand, Birjand, Iran

*Correspondence to*: Pei-Ling Wang (plwang@ntu.edu.tw)

**Abstract.** Terrestrial mud volcanoes (MVs) represent the surface expression of conduits tapping fluid and gas reservoirs in the deep subsurface. Such plumbing channels provide a direct, effective means to extract deep microbial communities fueled by geologically produced gases and fluids. The drivers accounting for the diversities and compositions of these MV microbial communities distributed over a wide geographic range remain elusive. This study characterized the variation of microbial communities in 15 terrestrial MVs across a distance of ~10,000 km of the Eurasian continent to test the validity of distance control and physiochemical factors in explaining biogeographic patterns. Our analyses yielded diverse community compositions with a total of 28,928 amplicon sequence variances (ASVs) taxonomically assigned to 73 phyla. While no true cosmopolitan member was found, ~85% of ASVs were confined within a single MV. Community variance between MVs appeared to be higher and more stochastically controlled than within MVs, generating a slope of distance–decay relationship exceeding those for marine seeps and MVs, and seawater columns. For comparison, physiochemical parameters explained 12% of community variance, with chloride concentration being the most influential factor. Overall, the apparent lack of fluid exchange renders terrestrial MVs a patchy habitat, with microbiomes diverging stochastically with distance and consisting dispersal-limited colonists that are highly adapted to the local environmental context.

## 1 Introduction

Microbial biogeography describes the distribution of microbial taxa over space and time, providing insights into the fundamental processes generating and governing diversity (Lomolino et al., 2006). Four nonexclusive processes—selection, drift, dispersal, and mutation—have been proposed to account for various microbial biogeographical patterns (Hanson et al., 2012). The "selection" process is generally regarded as deterministic and involves nonrandom, niche-based mechanisms,

including environmental filtering (e.g., pH, temperature, salinity, and geochemical or redox variations) and various biological interactions (e.g., competition, mutualisms, predation, and tradeoffs) (Ning et al., 2019). Therefore, the distribution pattern of

microbial diversity is controlled by the response of community members to environmental parameters (Comte et al., 2016; Power et al., 2018). Such selection factors could have led to the establishment of a variety of core microbiomes inhabiting distinct environments, such as soil, sediment, aquatic, and vent ecosystems (Orcutt et al., 2011; Ruff et al., 2015) or organized spatially as in a gradient and, thus, autocorrelated (Hanson et al., 2012; Ranjard et al., 2013). The "drift" process is caused by chance events (e.g., differences of taxa associated with birth and death events), differentiating microbial composition over

space in neutral theory (Slatkin, 1993; Condit et al., 2002). Microbial dispersal is defined as the physical movement of cells between two locations and successful establishment at the receiving location (Hanson et al., 2012). Due to the dispersal limitation, chance events at one location would influence nearby compositions. Therefore, the interaction between drift and dispersal limitation would generate a distance–decay relationship (DDR) (Hutchison and Templeton, 1999) in which the community dissimilarity increases with distance. Finally, gene duplications, mutations, and other processes produce new genes

and alleles that reshape the DDR by increasing local genetic diversity across all locations. Although these latter three processes generate community diversity patterns indistinguishable from random chance alone or are considered a stochastic consequence (Ning et al., 2019), they are also non-exclusive and interact with deterministic processes. Dissecting the contribution of individual processes and governing factors remains a challenging issue (Hanson et al., 2012).

Mud volcanoes (MVs) represent a unique ecosystem for investigating microbial biogeographic patterns when compared with

aquatic, soil, and sediment ecosystems on or near the land surface or seafloor. This uniqueness stems from the fact that the MV's genesis is tightly linked with the plumbing of fluids and sediments from deep reservoirs through fracture networks often extending to a depth of several kilometers (Mazzini and Etiope, 2017). Because advection dominates over diffusion for fluid transport, relatively rapid migration can occur with minimal alterations of geochemical characteristics and even microbial communities of fluids/muds emanating from a mud cone or pool (Dimitrov, 2002; Chen et al., 2020). Therefore, MVs provide

a direct, effective means to recover deep microbial communities. Meanwhile, the export of reducing compounds and rapid deposition of sediments enable MV sediments to be highly reduced and confined to a limited spatial extent (from tens of cms to kilometers) (Chang et al., 2012; Cheng et al., 2012; Mazzini and Etiope, 2017) Such physico–chemical characteristics generate localized, strong redox gradients and host abundant microorganisms with identities distinct from adjacent environments or overlying seawater (Wang et al., 2014; Lin et al., 2018), rendering MVs globally distributed, unique biological

hotspots fueled by geologically produced gases and fluids. A recent survey has demonstrated the predominance of few cosmopolitan taxa with a physiological preference for methane or hydrocarbons in marine MVs (Ruff et al., 2015). This line of evidence, combined with the observed DDR, suggests that both dispersal and selection exert a profound influence on shaping community compositions and structures. In contrast to the aid of dispersal through seawater circulation in marine counterparts, terrestrial MVs are even more limitedly connected between each other. For a geographic scale larger than tens of kilometers,

dispersal through groundwater transport would be essentially absent. This limitation, combined with enormous oxidative power driven by atmospheric oxygen (Lin et al., 2018) renders the terrestrial MVs ideal for investigating whether any biogeographic

pattern imposed by geographic isolation and environmental contexts emerges. In addition to various spatial scales, environmental and redox contexts vary substantially along a vertical scale. The variance of beta diversity and its controlling mechanism on both spatial and vertical scales remains poorly constrained.

This study aims to determine prokaryotic community compositions and structures associated with terrestrial MVs and to constrain the underlying mechanisms by examining the control of deterministic and stochastic processes on community variations over a spatial scale (up to ~ 10,000 km) across the Eurasian continent and a vertical scale (up to ~ 1.5 m) over a redox transition. Community compositions based on 16S rRNA gene sequences and metadata for cored sediments from 15 MVs were synthesized and analyzed to assess the control of qualitative stochasticity on community variations at different

spatial and vertical scales using various statistical approaches and ecological metrics. Moreover, while cosmopolitan members with significant dispersal capability and specific colonists highly adapted to local environmental contexts were identified, distribution patterns of members potentially possessing sulfur or methane metabolisms were also revealed. These results were compared with marine data to draw the framework and characteristics shared between terrestrial and marine MV ecosystems. This work represents the most extensive microbial ecology study to date on terrestrial MVs at a continental scale.

**2 Materials and Methods**

**2.1 Sampled MVs and data source**

Muddy fluids from bubbling pools and sediment cores from the adjacent mud platform were retrieved from MVs across the Eurasian continent during 2011 to 2013 (Fig. 1; Table S1) for geochemical (n=9) and molecular analyses (n=13). Detailed sample collection, processing, and preservation are described in the supplementary information. Data obtained in this study

were merged with companion geochemical data for 4 MVs in Italy (AR01, COM01, PA01, and PA02; Chiu, 2015), and geochemical and molecular data for 2 MVs in Taiwan (LGH03 and SYNH02; Tu et al., 2017; Lin et al., 2018) to generate a total of 136 sample sets for 16 cores from 15 MVs.

**2.2 Geochemical analyses**

Concentrations of methane were analyzed using a 6890N gas chromatograph (GC; Agilent Technologies, USA). Carbon

isotope compositions of methane were measured with a MAT253 isotope ratio mass spectrometer connected to a GC Isolink (Thermo Fisher Scientific, USA). Chloride and sulfate concentrations in porewater were analyzed using an ICS-3000 ion chromatograph (Thermo Fisher Scientific, USA). Concentrations of particulate total organic carbon (TOC), total inorganic

carbon (TIC), total nitrogen (TN), and total sulfur (TS) were determined by an elemental analyzer (MICROcube, Elementar, Germany). Detailed methods for these analyses are described in the supplementary information.

## 2.3 Microbial community analyses

Crude DNA for 16S rRNA gene analyses was extracted from fluids/sediments using a PowerSoil DNA Isolation Kit (Qiagen, Germany). Bubbling fluids (if available) and sediments distributed across geochemical transition were selected for DNA extraction. These samples are representative of communities inhabiting the subsurface source region (for bubbling fluids) or subject to the redox gradient developed after the sediment deposition (cored sediments in the adjacent mud platform). DNA extracts were obtained and stored at −80 °C for subsequent analyses.

Amplicons for 16S rRNA genes were generated from polymerase chain reactions (PCR) using the universal primers targeting both bacterial and archaeal communities, and sequenced on the Illumina platform. Sequences were analyzed using Mothur and QIIME2 (Schloss et al., 2009; Bolyen et al., 2018). Denoised reads were assembled to full sequences, aligned, and taxonomically assigned against the Silva v.132 reference set using Mothur. Detailed schemes for PCR, sequencing, and sequence processing are described in the supplementary information. The obtained sequences were deposited in GenBank with accession number PRJNA560274.

## 2.4 Statistic analyses

Detailed methods for statistical analyses are described in the supplementary information.

### 2.4.1 Microbial community analyses

Sequence data were first rarefied to 9,413 sequences per sample through 100 sequence random re-sampling (without replacement) of the original amplicon sequence variant (ASV) table to account for the difference in sequencing depth for the calculation of alpha diversity indices (Hill, 1973). For the beta diversity calculation, the entire ASV table was used and normalized using the function cumNorm embedded within metagenomeSeq in R (Paulson et al., 2013). The method considers the sum of ASVs and their quantile distribution for each sample by adjusting the sequence number for individual ASVs while keeping the total ASVs the same before and after the normalization. The method does not sacrifice the sample diversity contributed from rare ASVs, thereby providing a better assessment of the community variation across different spatial scales or controlled by localized environmental factors. The dissimilarity matrix between samples was computed using the Bray-Curtis method (Bray and Curtis, 1957; Ranjard et al., 2013) and visualized through the ordination of non-metric multidimensional scaling (NMDS). Among the synthesized 136 samples, 126 samples with concentrations of chloride, sulfate,

methane, TN, TS, TIC, and TOC were used for constrained correspondence analysis (CCA), which aims to elucidate the relationship between microbial community compositions and geochemical variables.

### 2.4.2 Habitat similarities

Habitat similarities were calculated from the Euclidean distances between paired 126 samples with the available concentrations
of chloride, sulfate, methane, TN, TS, TIC, and TOC using the following equation (Ranjard et al., 2013):

$$E_d = \left(1 - \frac{\mathrm{Euc}_d}{\mathrm{Euc}_{max}}\right) \tag{1}$$

where $E_d$ is the habitat similarity, $\mathrm{Euc}_d$ is the Euclidean distance, and $\mathrm{Euc}_{max}$ is the maximum distance between MVs.

**2.4.3 Distance decay relationships (DDR)**

To assess the DDR, pairwise community similarities were calculated using the Sørensen–Dice index (Dice, 1945). The pairwise similarity was transformed in a logarithmic scale to enhance the linear fitting (Nekola and White, 1999) using the following equation:

$$\log_{10}(S_{com}) = \log_{10}(a) + \beta \log_{10}(D) \tag{2}$$

where $S_{com}$ is the pairwise similarity in community composition, $D$ is the distance between two samples, $a$ is the intercept, and $\beta$ is the slope. The distance between samples was aggregated from two categories for samples in separate cores (geographic distance) or within the same cores (vertical distance).

**2.4.4 Normalized stochasticity ratios (NSTs)**

NSTs were calculated to assess the stochasticity of community variations within each category of samples using 50% as a threshold for either more stochastic (>50%) or deterministic (<50%) control (Ning et al., 2019). The analysis was conducted using the obtained ASV table by first categorizing all samples into "bubbling fluid", "surface sediment", and "within MV sediment", each representing source communities, source communities subject to minor surface impact, or communities
potentially altered by localized geochemical/redox context. The categorization addressed the scenarios whether the variation pattern for source communities with/without minor impact of the surface process (for "bubbling fluid" and "surface sediment") is dependent on distance separation and whether the variation pattern for localized communities (for "within MV sediment")

is dependent on a specific environmental context within individual MVs. The NST was calculated using the package NST developed by Ning et al. (2019) based on the Bray–Curtis dissimilarity ($NST_{bray}$), Jaccard distance ($NST_{Jaccard}$), and phylogenetic distance (pNST) between pairwise communities within each category of samples, following the recommendation of the developers.

## 3 Results

### 3.1 Physical and geochemical characteristics

The pairwise distance between samples ranged from 2.5 to 160 cm within cores and 0.005 to 9,924 km between cores (Fig. 1). Geochemical profiles of pore water showed various characteristics related to abiotic and microbial processes. Chloride concentrations varied highly among MVs (ranging between 82 mM at SI02 in Myanmar and 4890 mM at GG01 in Iran) and generally decreased with increasing depth within individual cores (Fig. S1). Sulfate concentrations ranged from below the detectable level at SM22, AK03, GJ01, TA, PA01, PA02, and LGH03 to 288 mM at GG01, with most data clustering between 0.5 and 2 mM (Fig. S1). Methane concentrations ranged between 0.006 mM (PA02) and 3.98 mM (SYMH02C4), with most data clustering between 0.2 and 1 mM (Fig. S2). The $\delta^{13}C$ values of methane spanned from -58‰ to -35‰ and exhibited a trend opposite to that of methane concentration. The molar ratios of methane over ethane and propane (C1 (methane) / (C2 (ethane) + C3 (propane))) were variable and ranged from 22 (SI02) to approximately 1200 (AR01 and COM01; Fig. S3). Detailed porewater characteristics are provided in the supplementary information.

Pairwise comparisons between samples within individual cores yielded a habitat similarity (*Ed*) ranging between 0.42 and 1.0 (for data points with a distance of 2.5 to 160 cm in Fig. 2a). These narrow-ranging indices were generally higher than those for samples between cores (for data points with a distance greater than 160 cm in Fig. 2a). Inspection of the data sets, however, demonstrated a contrast pattern between some MVs. For example, the geographic distances between MVs in Italy and Taiwan were the highest among all MV pairs. Habitat similarities between AR01/COM01 and SYNH02/LGH03 were greater than 0.96. In contrast, habitat similarities between GG01 and other MVs were low, even though the geographic distances were short. Overall, habitat similarities were not significantly correlated with vertical distance but horizontal distance ($P < 0.001$) (Fig. 2a).

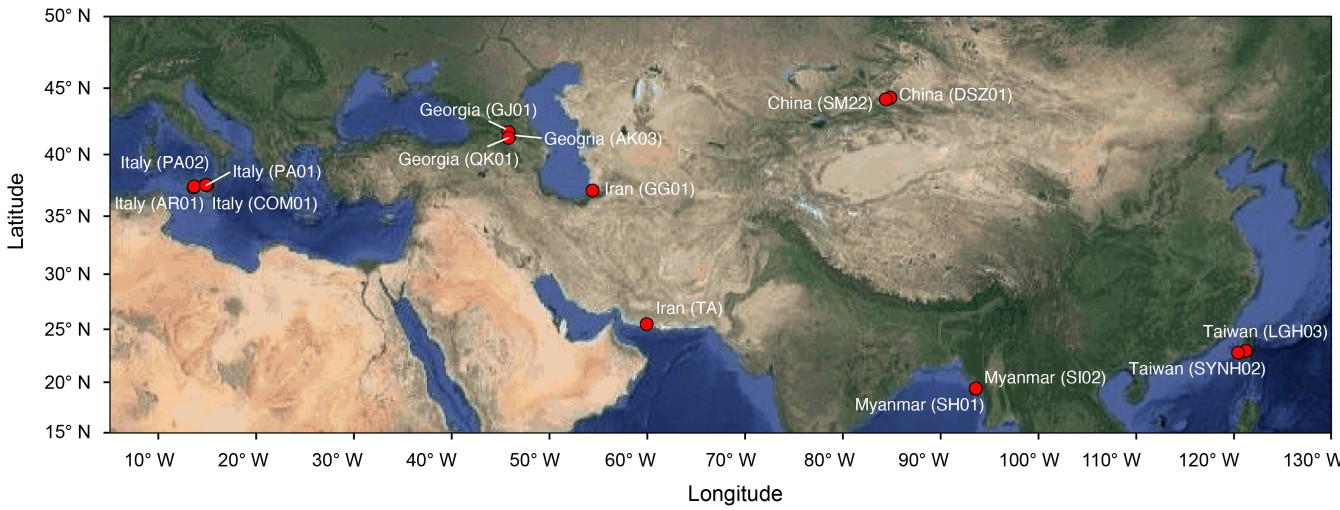

**Figure 1: Map overlay with analyzed MVs (solid red circles). Country names are shown with MV codes in parentheses. The map is modified from Google Maps 2021 using the ggmap package (Kahle and Wickham, 2013) in R.**

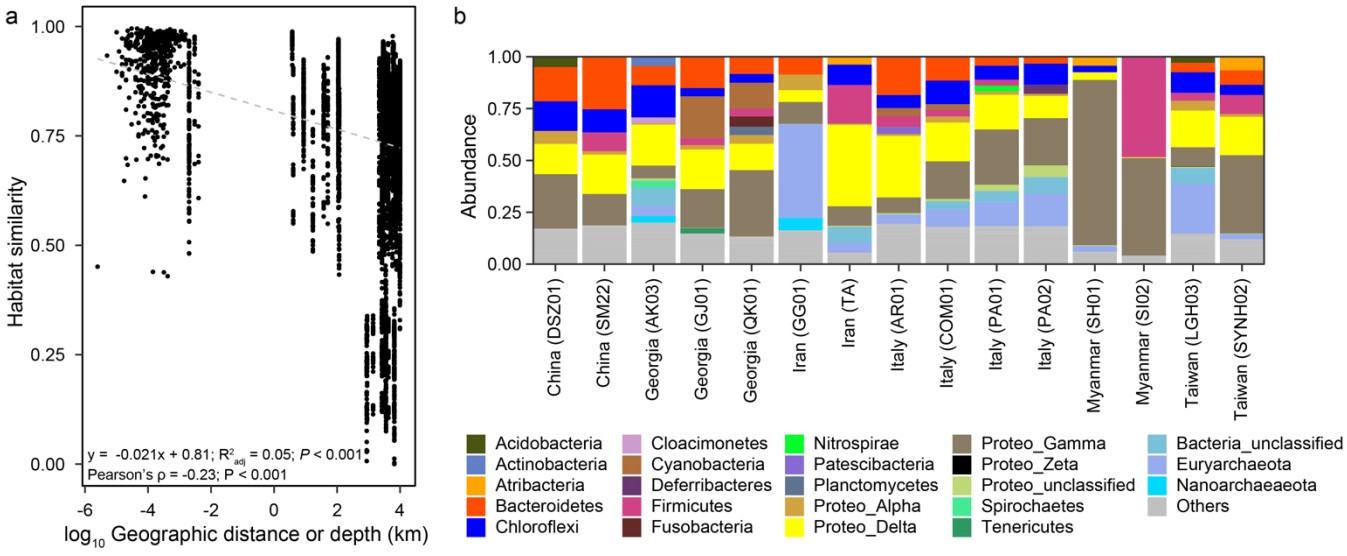

**Figure 2: (a) Plot of habitat similarity versus geographic distance or depth. (b) Abundances of major phyla based on 16S rRNA gene amplicons. Raw sequence data for LGH03 and SYNH02 from Tu et al. (2017) and Lin et al. (2018) were incorporated for the analysis and comparison.**

## 3.2 Community structures and compositions

A total of 24,617 bacterial and 4,311 archaeal ASVs, representing 181 classes (157 bacterial and 24 archaeal) within 73 phyla, were recovered. The observed ASVs for individual samples ranged between 58 and 1,462, with an average value of $449 \pm 250$ when singletons (presence of one sequence for an ASV at only one depth) were included. The trends of diversity indices were revealed in a similar pattern (Fig. S4). The lowest values of alpha diversity indices occurred for SI02 and SH01 in Myanmar, whereas the highest values were found for AR01 in Italy. Diversities at the ASV level were fully captured for individual cores but not sufficiently recovered by taking all cores as a whole (Supplementary information; Fig. S5).

The dominant phyla and subdivisions of Proteobacteria (>5% of the total reads) included Firmicutes (6.0%), Chloroflexi (7.6%), Euryarchaeota (8.6%), Bacteroidetes (9.5%), Deltaproteobacteria (18.6%), and Gammaproteobacteria (21.4%). The majority of bacterial reads were assigned to the orders Betaproteobacteriales (4.2%), Desulfuromonadales (6.1%), and Desulfobacterales (7.1%). Most sequences belonging to these three orders were related to the families Hydrogenophilaceae (2.9%), Desulfuromonadaceae (3.4%), and Desulfobulbaceae (4.5%), respectively. The two dominant genera, *Thiobacillius* (Hydrogenophilaceae) and *Desulfurivibrio* (Desulfobulbaceae), constituted 2.9% and 3.2% of the total reads, respectively. For comparison, the majority of archaeal reads were assigned to the orders Halobacteriales (2.8%) within Halobacteria and Methanosarcinales (4.3%) within Methanomicrobia. Whereas most sequences assigned to Halobacteria were related to the families Halobacteriaceae (1%) and Haloferacaceae (1.1%), the predominant sequences within Methanomicrobia were affiliated with ANME-2a (3.2%). Among the 28,928 ASVs, five out of the ten most abundant ASVs were affiliated with the genus *Thiobacillus* (0.4–0.7% of the total reads).

Of 73 phyla obtained, nine were found in all cores and the other nine in only one core (Figs. 2b & 3a). Cosmopolitan phyla were more abundant than endemic ones, with the exceptions for GG01, SYNH02, and SH01. Proteobacteria appears to be the only phylum present in all 136 samples (and all MVs) and the most abundant phylum in nearly all MVs (except for GG01 and SI02). Whereas the remaining eight phyla were present in all MVs, they were absent in few samples and occurred in 124 to 135 samples. The proportions of cosmopolitan taxonomic units decreased from the level of phylum to ASV (Fig. 3b – 3e).

Among the detected 1,214 genera, the genus *Desulfurivibrio* was the most abundant one and present in 98 of the 136 samples. Two other prevalent genera were detected in 126 samples and taxonomically assigned to the unclassified genera related to Anaerolineae within Chlorofexi and to Gammaproteobacteria, respectively.

At the ASV level, no truly cosmopolitan ASVs were identified (Fig. 3f). Pairwise comparisons yielded a shared 0–15.4% and 0–51.3% of the ASVs between MVs and between samples, respectively. In particular, the communities at SI02 completely differed from AK03, SM22, and LGH03. The most widely distributed ASVs were present in 9 MVs (Fig. 3f & Table S2) and constituted only 0.4% of the total sequences. Their sequences were affiliated with either the unclassified genus of Desulfuromonadaceae or *Desulfotignum*. Compared with the pattern based on higher taxonomic units (between genus and class), the ASVs confined at one MV and less than/equal to three MVs constituted 85% and 99% of the total ASVs, respectively, and were taxonomically diverse and unevenly abundant. Detailed information for the 10 most abundant ASVs is given in Table S2.

Within individual cores, the number of phyla ranged from 12 (SI02) to 62 (PA01). Although 16.1% (PA01)–58.5% (AR01) of detected phyla were shared between samples within individual cores, 6.3–40% of phyla were restricted to single samples. Similar to the pattern for phyla, the lowest number of genera occurred for SI02 (52), whereas the highest number was present for PA01 (550). In addition, 3.4% (GG01) to 26.5% (AR01) of genera, and up to 13.2% (SI02) of ASVs were shared within individual cores. In contrast, up to 49.9% (PA02) of genera and 83.9% (GG01) of ASVs were restricted to a single sample. Overall, community dissimilarities appear to be more pronounced between samples from distinct MVs than from within the individual MVs, indicating a high degree of endemism (Fig. 4a).

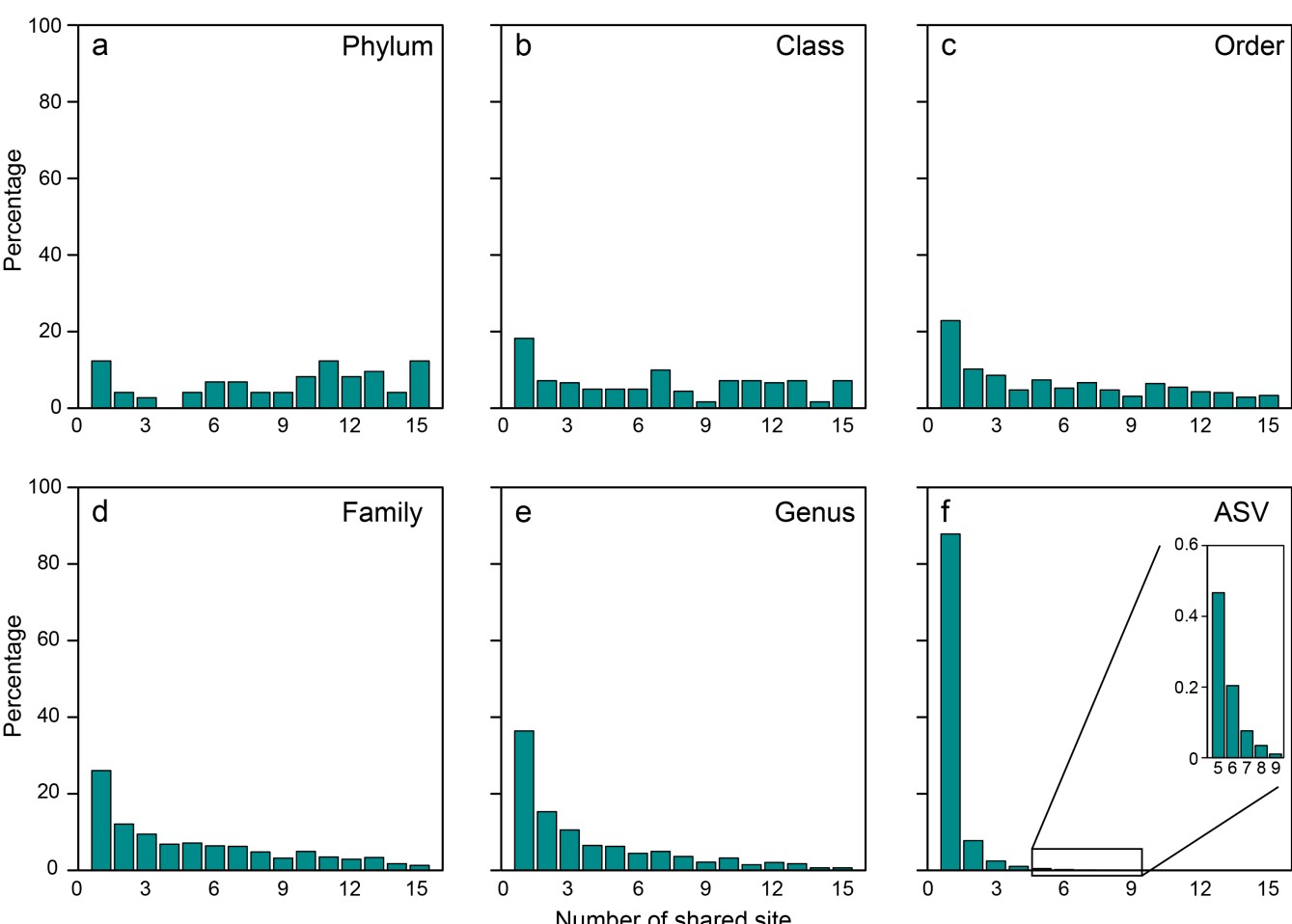

**Figure 3: Proportions of specific taxonomical units shared between the number of MVs. (a) phylum, (b) class, (c) order, (d) family, (e) genus, and (f) ASV.**

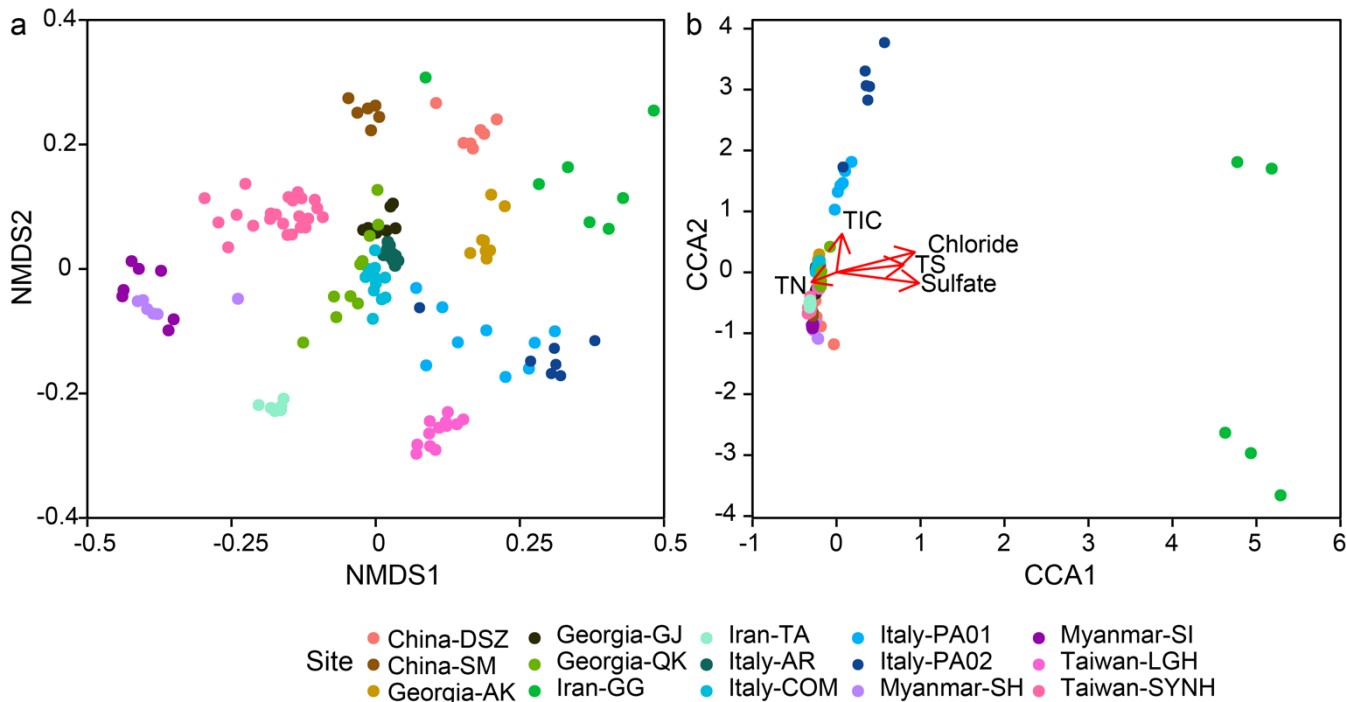

**Figure 4: Variance of 16S rRNA community compositions deduced from (a) the NMDS analysis based on the Bray–Curtis distance and (b) the CCA analysis based on the Chi-square distance. Ordination of significant geochemical parameters was overlayed for comparison in (b).**

### 3.3 Environmental effects

Multiple regression analysis yielded that methane, TN, and TIC concentrations had meaningful contributions (summed to be 18.8%) to the Shannon index (Table S3), with TIC being the most influential one (13.4% linear regression, $P < 0.001$; Table S4). Similarly, the TIC concentration was also significantly correlated with the Shannon index (Pearson's coefficient: $|r| = 0.38$, $P < 0.001$; Fig. S6).

Community dissimilarities were correlated more strongly with chloride (Mantel: $\rho = 0.45$, $P < 0.001$) than with sulfate (Mantel: $\rho = 0.26$, $P < 0.001$) concentrations and other geochemical parameters (Table S5). Permutational multivariate analysis of variance in community assemblage showed that TIC (3.6%) and TN (3.1%) concentrations had the highest contribution to the beta diversity, followed by sulfate (2.9%) and chloride (2.6%), TOC (2.4%), TS (2.0%), and methane concentrations (0.9%) ($P < 0.001$; Table S6).

The CCA yielded that the eight environmental parameters combined (sample depth, and concentrations of chloride, sulfate, and methane, TIC, TOC, TN, and TS) explained 12% of community dissimilarity (Fig. 4b). Of these factors, chloride, sulfate, TIC, TS, and TN significantly contributed to the overall differences in community composition. For communities within individual cores, a combination of various factors described above was significantly correlated with community dissimilarities

(Fig. S7). For example, the depth factor was significant for the community dissimilarities within 11 out of 16 individual cores. In contrast, the chloride factor was only significant for the community dissimilarities within two individual cores (QK01 and SYNH02C4). Finally, none of the selected factors were significantly correlated with the community dissimilarities within TA01, SH01, SI01, SM22, and LGH03.

### 3.4 DDR and NST patterns

Community structure varied widely across a scale of 9,924 km (Bray–Curtis $R_{ANOSIM} = 0.967$, $P < 0.001$; Fig. S8). Both the proportions of shared ASVs and community similarities significantly decreased with increasing geographical distance (Mantel: $\rho = -0.80$, $P < 0.001$), indicating a significant DDR with a slope coefficient, $|\beta|$, of 0.226 ($P < 0.001$) (Figs. 5a & b). If only communities from individual cores were considered, a DDR (Mantel: $\rho = 0.65$, $P < 0.001$) with an even higher $|\beta|$ value of 0.241 was obtained (Fig. S9b). Close examinations revealed that similarities of communities distributed within individual cores
were not significantly correlated with distance (or depth; Figs. S9d & e). Such variations in DDR reveal that community compositions were controlled by different mechanisms at vertical versus horizontal scales. Furthermore, the $NST_{bray}$ values varied from 100% for "bubbling fluid", 100% for "surface sediment", to between 0.89% and 62.54% for "within MV sediment" (Table S1). For "within MV sediment" category, the $NST_{bray}$ values for 12 out of 15 MVs were less than 50% (Table S1). For comparison, the same $NST_{Jaccard}$ values for "bubbling fluid" and "surface sediment", and a similar $NST_{Jaccard}$ range for "within
260 MV sediment" (1.01% – 60.90%) were obtained (Table S1). The pattern for pNST values resembled those for $NST_{bray}$ and $NST_{Jaccard}$ values (Supplementary information; Table S1).

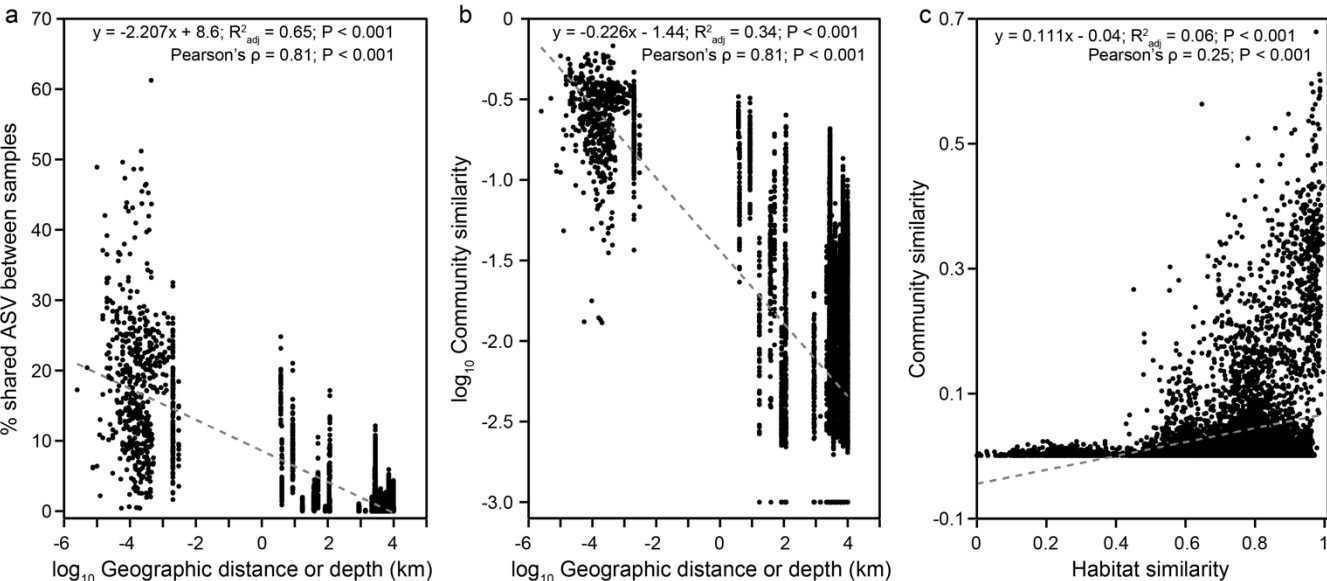

**Figure 5: Biogeographic patterns for MV microbial communities. Plots of (a) proportion of shared ASV and (b) community similarity versus geographic distance or depth, and (c) community similarity versus habitat similarity based on pairwise comparison for a total of 126 samples.**

## 4 Discussion

### 4.1 Environmental selection in terrestrial mud volcanoes

This study recovered diverse and complexly structured microbial communities in terrestrial MVs distributed across the Eurasian continent. Further linear regression analyses yielded that the Shannon index was mostly controlled by TIC (Fig. S6 and Tables S3–S4). Both the Mantel test and CCA revealed that community similarities were strongly influenced by chloride, sulfate, TIC, and TS concentrations (Fig. 4b and Table S5) and correlated with habitat similarities (Fig. 5c and Fig. S10). To explore the controlling factors for community variance further, the individual factors stated above were addressed at the community level first, followed by at the level of specific, abundant lineage.

Salinity has been identified as the primary factor shaping the distribution of microorganisms (Or et al., 2007). Because chloride is inert to most catabolic and abiotic reactions, any correlation between community (Or et al., 2007) index and chloride concentration could have reflected the physiological response or tolerance to salinity variation (Or et al., 2007) related to evaporation and evaporite dissolution/precipitation. Chloride concentrations measured in this study spanned over a broad range between 82 and 4890 mM. Whereas the community similarity was shaped by chloride (Fig. 4b and Supplementary Table S5), the abundances of specific lineages, including Haloferacaceae and Halobacteriaceae within Halobacteria, were also significantly correlated with the chloride concentrations (Fig. S10). In particular, the order Halobacteria detected in 11 out of the 15 investigated MVs was highly enriched in hypersaline MVs (GG01, PA01, and PA02) where the chloride concentrations reached up to 4890 mM. Previous culture tests have shown that strains affiliated with this lineage can cope with the stress imposed by high osmotic pressures and low water activities (Grant, 2004). All members of the family Halobacteriaceae grow optimally at chloride concentrations of above 2500 mM (Oren, 2014) and up to 5000 mM (*Halobacterium* sp. NRC-1) (DasSarma and DasSarma, 2001). In contrast, the abundances of JS1 within Atribacteria and Hydrogenophilaceae within Proteobacteria were negatively correlated with the chloride concentrations ($\rho < -0.5$; Fig. S10). Although these two families were prevalently distributed in all MVs, the correlation pattern suggests their sensitive response to the salinity stress and preference for low salt conditions.

TIC in sedimentary systems represents a pool of biological carbonate remains and authigenic carbonate formed or induced by microbial processes (Zheng et al., 2011) Considering that carbonate fossils are exempted from the preservation of genetic materials during burial diagenesis (Allison and Pye, 1994), the correlation between TIC and community similarity could have been controlled by the composition of heterotrophs and methanotrophs capable of converting organic carbon and methane into dissolved carbonate and eventually to the precipitation of carbonate minerals. Therefore, the presence and concentration of TIC might reflect microbial capability for the utilization of organic carbon or methane to some degrees. In addition to the

community variance, the abundances of a variety of families, such as Woesearchaeia, Thiohalorhabdaceae, Marinilabiliaceae, Lentimicrobiaceae, Haloferaceae, Halobacteriaceae, Ectothiorhodospiraceae, Desulfarculaceae, Balneolaceae, and Anaerolineaceae were found to be significantly correlated with the TIC concentrations (Fig. S10). Whereas none of the methanotrophs are related to the lineages described above, previous studies have demonstrated that a large number of strains affiliated with Halobacteriales, Marinilabiliaceae, Lentimicrobiaceae, Desulfarculales, and Anaerolineale are capable of metabolizing various forms of organic carbon (e.g., fatty acids, sugars, amino acids, hydrocarbons, and even short-chain alkanes) (Yamada and Sekiguchi, 2009; McGenity, 2010; Kuever, 2014; McIlroy and Nielsen, 2014; Borrel et al., 2019; Mori et al., 2019). Moreover, metagenomes with 16S rRNA gene sequences affiliated with Woesearchaeota contain genes for starch/sugar utilization, glycolysis, folate C1 metabolism, and fermentation (Liu et al., 2018). The gene pattern further suggests the heterotrophic nature of Woesearchaeota and its potential requirement of metabolic complement from other microorganisms (e.g., acetate-utilizing methanogens). Overall, the physiological characteristics derived from previous cultivation experiments, along with metagenomic data, all demonstrate the prevalence of heterotrophy among these phyla/orders. The positive correlation between their abundances and TIC concentration suggests a connection between carbon utilization and carbonate precipitation. We noted that a similar pattern was not observed for TOC and TN (Table S5). It is likely that the pools of bioavailable and biodegradable TOC and TN only constitute a small fraction of the pool size, thereby rendering TOC and TN less sensitive to the community variance.

Compared with chloride, sulfate concentrations varied at a higher magnitude (the coefficient of variation) of 13–186%; Table S7). With the exception of SH01 and SYNH02, the sulfate concentrations were not significantly correlated with the chloride concentrations (Table S7). The decoupling of sulfate from chloride suggests that in addition to the evaporation or dissolution/precipitation of evaporite minerals, microbially mediated sulfate reduction or oxidation of reduced sulfur plays a role in controlling sulfate abundance. Whereas the sulfate concentrations were significantly correlated with the community similarities (Table S5), abundant sulfur oxidizers and sulfate reducers (such as Hydrogenophilaceae-, Desulfobulbaceae-, and Desulfuromonadaceae-related members; Or et al., 2007) were detected at SI02, SH01, and AR01. In addition, the abundances of sulfur-metabolizing lineages, such as sulfur-oxidizing *Thiobacillus* members within Hydrogenophilaceae ($\rho = 0.38$, $P < 0.001$) and sulfate-reducing members related to Desulfobulbaceae and Desulfarculaceae ($\rho = -0.40$, $P < 0.001$; Fig. S10), were found to be significantly correlated with the sulfate concentrations.

Similar to TIC, TS represents an aggregation of various sulfur-bearing minerals formed through different processes at varying time scales. These pools of minerals include pyrite (or other sulfide minerals) and gypsum precipitated over geological time and sulfide minerals (e.g., iron monosulfide and pyrite) produced from microbial sulfate reduction at a contemporary time scale (Halevy et al., 2012). In contrast to marine environments where the sulfate pool is enormous, terrestrial MVs are often devoid of sulfate, unless evaporite is ubiquitous. Therefore, in situ sulfate reduction proceeds with sulfate produced from microbial sulfur oxidation or gypsum dissolution (Canfield, 1989; Yao and Millero, 1996; Weber et al., 2017). In this regard, the correlation between TS and community similarity observed in this study demonstrated that in situ microbial processes played a role in shaping community compositions. Detailed analyses further revealed that the abundances of a variety of

families, such as Thiohalorhabdaaceae, Balneolaceae, and Haloferaceae, were positively correlated with the TS concentrations (Supplementary Figure S10). Among these families, most strains affiliated with Thiohalorhabdaceae can directly metabolize sulfur (Sorokin et al., 2020). In contrast, the abundances of lineages, such as Methanosaetaceae, Marinobacteraceae, Methylomonaceae, and JS1, were negatively correlated with the TS concentrations. Although these lineages have been commonly observed at the sulfate-to-methane transition in marine sediments (Orphan et al., 2001; Inagaki et al., 2006), the correlation pattern suggests their proliferation in sulfur-depleted environments.

Methane has been found to be abundant in most MVs (Etiope et al., 2019), providing an energetic substrate and carbon source for various metabolisms. Its abundances varied substantially (CV of 277%) and were neither correlated with the chloride concentrations nor community similarities. Previous studies have demonstrated that carbon isotopic compositions and C1 / (C2 + C3) abundance ratios could be used to distinguish methane produced from methanogenesis from thermal maturation (Whiticar, 1999). Thermogenic hydrocarbon gases generally possess C1 / (C2 + C3) abundance ratios ranging from 0 to 50 and carbon isotopic compositions of methane greater than -50‰ (Claypool and Kvenvolden, 1983), whereas microbial sources generate hydrocarbons with C1 / (C2 + C3) abundance ratios generally greater than 1000 and carbon isotopic compositions of methane smaller than -60‰ (Claypool and Kvenvolden, 1983). Regardless of the production source, microbial methane consumption would impart carbon isotopes of methane, preferentially producing $^{12}$C-enriched $CO_2$ or leaving residue methane enriched with $^{13}$C.

In this study, rather than community dissimilarity, the abundances of methanogens (members of Methanosaetaceae) were significantly correlated with the methane concentrations ($\rho$ = 0.23, $P$ < 0.001; Fig. S10). Although both C1 / (C2 + C3) abundance ratios and carbon isotopic compositions of methane revealed a mixed origin of methane (Figs. S2 & S3), the correlation pattern supports a quantitative role of microbial over thermogenic methane production in terrestrial MVs. Furthermore, both aerobic and anaerobic methanotrophs were detected in 11 of the 15 investigated MVs. Whereas these two types of methanotrophs possess contrasting oxygen affinities, they were all distributed from the surface to the bottom of investigated cores. Resembling the findings in the marine setting (Ruff et al., 2015), neither the abundances of the entire ANME (Fig. S10) nor the community dissimilarities were significantly correlated with the methane concentrations. The abundances of the whole ANME and ANME-2a/b were inversely correlated with the carbon isotopic compositions of methane and sulfate concentrations ($\rho$ = -0.37 and -0.32 for whole ANME, $P$ < 0.001; $\rho$ = -0.42 and -0.45 for ANME-2a/b, $P$ < 0.001), respectively, a pattern consistent with the coupling of methane oxidation with sulfate reduction mediated by ANME-2a/b and Deltaproteobacteria (Knittel and Boetius, 2009). In addition, ANME-2d related sequences were mostly distributed at LGH03. Previous culture and field studies have shown that ANME-2d related members can oxidize methane with the reduction of nitrate, iron, and manganese (Beal et al., 2009; Haroon et al., 2013; Ettwig et al., 2016; Scheller et al., 2016). Our findings suggest that anaerobic methanotrophy driven by electron acceptors other than sulfate is not prevalent in terrestrial MVs.

Finally, the factor of depth represents an integration of geochemical variations (e.g., sulfate, methane, and chloride). The counteraction between the downward penetration of atmospheric oxygen and upward migration of reducing methane would presumably result in a steep redox gradient along the depth (Lin et al., 2018), leading to a segregation of distinct niches with

community compositions adapted to various redox and geochemical affinities. Indeed, the within-core community similarity

was significantly correlated with depth for 11 cores (Fig. S7) and with habitat similarity for all cores (Fig. S9), a pattern consistent with what has been reported for marine sediments (Jørgensen et al., 2012; Ruff et al., 2015; Petro et al., 2017).

## 4.2 Microbial dispersal patterns

A distance–decay trend was identified for geographic distances up to ~10,000 km (Figs.1, 2, and 5). The deduced $|\beta|$ value (0.226) was smaller than that for macro-organisms ($|\beta|$ = 0.2–0.7) (Nekola and White, 1999), pointing to a higher dispersal

rate of microorganisms (Astorga et al., 2012; Zinger et al., 2014). Furthermore, the $|\beta|$ value resembled those for microbial communities in coastal sediments (Zinger et al., 2014) and was larger than those ($|\beta|$ = 0–0.15) for microbial communities in marine seeps and MVs (Ruff et al., 2015), and seawater columns (Zinger et al., 2014), suggesting a higher degree of dispersal limitation in terrestrial and transition zone sediments than in marine environments. Our results are in contrast to the biogeographic pattern derived from ammonia-oxidizing bacteria in salt marsh sediments where dispersal limitation at the local

scale contributes to the beta diversity, and no evidence of evolutionary diversification is observed at the continental scale (Martiny et al., 2011). Furthermore, the mean NST values for both "bubbling fluid" and "surface sediment" categories were high (100%) (Table S1). These sample categories represent either deeply sourced communities (for "bubbling fluid") or source communities susceptible to the alteration of surface processes (for "surface sediment"). Considering that most of these MVs are distant from each other (except for SI01 and SH01), fluid exchange and subsequently microbial dispersal or exchange

between MVs could be exempted. Therefore, the high NST values associated with these two categories suggest that highly stochastic control on community variance is likely linked to the limitation imposed by distance separation. In contrast, the mean NST value for "within MV sediment" category was 32%. The NST values were < 50% for 12 out of 15 MVs and between 59% and 63% for the remaining three MVs. This sample category represents a suite of sediments successively buried with depth by the mud emanating from bubbling pools or cones. With the counteraction of atmospheric oxidation and reducing

power derived from methane, sulfur, and organic matter, a strong redox and geochemical gradient develops as a result of the transient interaction between biogeochemical cycling and abiotic processes (Chang et al., 2012; Cheng et al., 2012; Wang et al., 2014; Tu et al., 2017; Lin et al., 2018). Therefore, the relatively low NST values suggest that the deterministic control on within-MV-community variance is predominantly impacted by the localized redox and geochemical context that is tightly linked to the channeling of mud and fluid sourced to the deep reservoirs.

Of diverse community members possessing key methane and sulfur metabolisms, ANME, Desulfobacterales, Methylococcales, and *Thiobacillus* were identified as the cosmopolitans, being present in 9–13 of the investigated MVs. The most abundant ANME ASV (accounting for 7.9% of all ANME sequences) appeared to be the most dominant one at LGH03 (2.2% of the total reads) but was present only as a few sequences at TA and PA01 (less than 0.1‰ at each MV). The most widespread ANME ASV was observed at 5 MVs in Italy and Georgia (Fig. S11), although it only represented 0.8% of all

ANME sequences. A similar pattern was observed for the most abundant ASVs of the Desulfobacterales and *Thiobacillus* that

represented 2% and 11% of individual lineages but were only present at 2 and 4 MVs, respectively. The most widespread ASVs of these lineages (Fig. S11) were observed at 9 MVs and only accounted for 0.3% of individual lineages. Comparisons with marine counterparts further revealed a higher degree of endemism for terrestrial MVs (Fig. 3; 88% and 70% of ASVs unique to one terrestrial and marine MV, respectively) (Kahle and Wickham, 2013) and drastically different community compositions mediating some of these key metabolisms. For example, sulfide-oxidizing Thiotrichales and methanotrophic ANME-1 and ANME-3 are prevalent in marine settings (Ruff et al., 2015). This finding is in contrast to the terrestrial cosmopolitan sulfur-oxidizing *Thiobacillus* and methanotrophic ANME-2a reported in this study.

Overall, the high $|\beta|$ and NTS values (Fig. 5), together with the high level of endemism in terrestrial MVs, reflect a strong pressure for local diversification. In terrestrial MVs, reduced materials are expelled to and distributed in a limited extent of the surface environment. A strong redox gradient is generated across a transect from the center of MVs to the surrounding region (Chang et al., 2012; Cheng et al., 2012; Wang et al., 2014; Tu et al., 2017; Lin et al., 2018). This phenomenon, combined with the lack of subsurface fluid exchange between terrestrial MVs, suggests that microbial dispersal is only facilitated through air circulation. Such a route imposed by strong oxidative power would be, however, particularly detrimental to further colonization of anaerobes in destination MVs. Exceptions analogous to the dispersal of thermophilic anaerobes from marine hydrothermal vents might occur if a protective agent, such as germination or sporulation, could be developed to cope with the stress associated with the exposure to the air (Bray and Curtis, 1957; Müller et al., 2014). The limitation in dispersal also renders terrestrial MVs a habitat patchily distributed and bears limited genetic exchange with the surrounding habitats.

## 5 Conclusions

We reported microbial community diversities and compositions associated with terrestrial MVs across the Eurasian continent. At a higher taxonomic level (phylum to order), a rather uniform composition of microbiomes was recovered from most MVs. The major phyla recovered included Proteobacteria, Chloroflexi, Euryarchaeota, Cyanobacteria, Firmicutes, Atribacteria, Bacteroidetes, and Actinobacteria. In contrast, abundant ASVs (a total of 28,928) were unevenly detected in various MVs, among which no true cosmopolitan ASV was found. Community similarities decreased and increased with geographic distances and habitat similarities, respectively. Although high NST values (100%) were observed for "bubbling fluid" and "surface sediment" communities, low NST values (< 50%) were derived for "within MV sediment" in 12 out of 15 MVs. The slope of the DDR was steeper than those for marine MVs, seeps, and water columns. Such community relatedness was significantly correlated with various physiochemical parameters, such as chloride, TIC, methane, and sulfate. Within individual cores, the significant correlation between community and habitat similarities highlights the importance of environmental filtering at a localized, vertical scale. In summary, the high $|\beta|$ and NST values combined with 85% of ASVs confined to individual MVs suggest the limit in microbial dispersal capability and a high degree of endemism. Such stochastic processes operating at continental scales in addition to deterministic filtering at local scales drive the formation of patchy habitats and the pattern of diversification in terrestrial MVs.

## Acknowledgments

We are grateful to the anonymous reviewers for their constructive and critical comments, and a number of scientists for their assistance in the field sampling. We want to offer special thanks to Prof. Sun-Lin Chung at Academia Sinica, Taiwan for his initiation and coordination of logistic arrangement for field works in Central Asia. This work was supported by the Ministry of Science and Technology (NSC 100-2627-M-002-003; MOST 109-2116-M-002-015; MOST 109-2116-M-110 -001 -MY3) and the Ministry of Education, Taiwan.

## Author contributions

LHL and PLW initiated and designed the study. YPC, LHL, LWW, FI, JBHS, and SNR performed field works. THT, LLC, YPC, LHL, LWW, and PLW performed laboratory and statistical analyses. All participated the discussion and paper writing.

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
