# Peer review of "Biogeographic pattern of microbial communities inhabiting terrestrial mud volcanoes across the Eurasian continent"

_Biogeosciences, 2021_

## Author Response (AR1)

**Reviewer 1**

**General comments:**

Tu et al. characterized microbial communities of 15 terrestrial MVs of the Eurasian continent to test the validity of distance control and physiochemical factors in explaining biogeographic patterns. Central to this manuscript is the claim that stochastic process determines the spatial variations in microbial communities inhabiting terrestrial mud volcanoes across the Eurasian continent. The experimental design is reasonable. Unfortunately, the study suffers many shortcomings:

**Comment 1:** The analysis of the article is not sufficient to support the conclusion drawn by the author. The authors emphasized stochastic process in the title, but they told nothing related to stochastic processes in either introduction or result or M&M section. How will readers know stochasticity and determinism if you do not mention it in the introduction? The author only analyzed the influence of environmental factors on microbial diversity and the distance attenuation relationship. How can you link these analyses to community assembly mechanisms? So, I don't know how the author concluded that random factors dominated the variation of microbial community.

**Response:** Thanks for the comment. We decide to change the title to a more general form as "Biogeographic pattern of microbial communities inhabiting in mud volcanoes across the Eurasian continent" to clear the inconsistency raised by the reviewer. Intrinsically, "drift", "dispersal", and "mutation" could all be categorized as stochastic processes even though they are non-exclusive and interact with deterministic processes. These processes would generate a community diversity pattern indistinguishable from random chance. In contrast, "selection" is regarded as a deterministic process which involves nonrandom, niche-based mechanisms, including environmental filtering (e.g., pH, temperature, salinity, and geochemical or redox variations) and various biological interactions (e.g., competition, mutualisms, predation, and tradeoffs). Such selection factors could have led to the establishment of a variety of core microbiomes inhabiting distinct environments or organized spatially as in a gradient and, thus, autocorrelated. We have revised the first paragraph of the "Introduction" to provide readers with more description about deterministic and stochastic processes. As being suggested by reviewer 2 and to provide the assessment of the relative stochasticity of community variance, we have introduced a new calculation of the NST index ("NST" stands for normalized stochastic ratio; Ning et al., 2019). We have categorized all samples into "bubbling fluid", "surface sediment", and "within site sediment", each representing source communities, source communities susceptible to minor surface impact, and communities potentially altered by localized geochemical context, respectively. The results demonstrate a high stochastic control on "bubbling fluid" (NST = 100%) and "surface sediment" (NST = 100%) and a relatively strong control of deterministic process on "within site sediment" (12 out of 15 MVs with NSTs < 50% and the remaining 3 with NSTs between 59% and 63%), a data pattern consistent with the results obtained from existing DDR, NMDS, and CCA analyses. Similar approach has been applied to the calculation of pNST, which is a measure of stochasticity based on phylogenetic distance between communities and represents an analogy of $\beta$NTI. The results demonstrate a pattern similar to that from NST. High stochastic control has been observed for "bubbling fluid" (pNST = 65%) and "surface sediment" (pNST = 73%), whereas mild deterministic control has been inferred for "within site sediment" (8 out of 15 MVs with pNSTs <

50% and the remaining 7 with pNSTs between 51% and 92%). Compared with ASV abundance dissimilarity, phylogenetic dissimilarity leads to a pattern with the stochastic control (pNST > 50%) on more MVs. This could be exemplified with MVs SI and SH where relatively small numbers of ASVs were recovered. The pNSTs for these two MVs are much higher than the corresponding NSTs (44% – 51% versus ~ 1%), suggesting that contrast proportions of phylogenetically distant ASVs between communities lead to a higher degree of random dissimilarity. While our existing analyses are primarily based on the ASV abundance dissimilarity (e.g., NMDS and DDR), we will present both the NSTs and pNSTs results.

**Comment 2:** The article lacks clear goals and scientific issues, which prevents readers to understanding the article quickly.

**Response:** Thanks for the comment. To make the goal even clearer, we have revised the objective of the study as "This study aims to determine prokaryotic community compositions and structures associated with terrestrial MVs and the underlying mechanisms by examining the control of deterministic and stochastic processes on community variations over a horizontal scale across the Eurasian continent and a vertical scale over a redox transition".

**Comment 3:** The author should check the article carefully before submitting it. There are many confusing descriptions and mistakes across the article.

**Response:** Thanks for the comment. We will revise the confusing descriptions more carefully.

**Specific comments:**

**Comment 1:** Lines 70-75: Clear goals and scientific issues are lacked.

**Response:** Thanks for the comment. To make the goal even clearer, we have revised the objective of the study as "This study aims to determine prokaryotic community compositions and structures associated with terrestrial MVs and the underlying mechanisms by examining the control of deterministic and stochastic processes on community variations over a horizontal scale across the Eurasian continent and a vertical scale over a redox transition".

**Comment 2:** Lines 80: There are 17 samples in Fig.S1, but 16 are written in the material.

**Response:** Thanks for the comment. Indeed, 16 cores were investigated in this study. We have revised the manuscript and supplementary information.

**Comment 3:** Lines 105: Clarify the number of ASVs.

**Response:** Thanks for the comment. Different sequencing depths could lead to the overestimate or underestimate of diversity (both alpha and beta diversities). To circumvent the shortcoming, ASV normalization using the function cumNorm embedded within metagenomeSeq in R was adopted in this study. The method considers the ASV quantile distribution for each sample by adjusting the sequence number for individual ASVs while keeping the total ASVs the same before and after the normalization. The method does not sacrifice sample diversity contributed from rare

ASVs, thereby providing a better assessment on the community variation across different spatial scales or controlled by localized environmental factors. Nevertheless, the number of ASVs in each sample has been added to Table S1.

**Comment 4:** Lines 110: Why not consider pH, which is generally believed to be the most important factor affecting the microbial community.

**Response:** Thanks for the comment. The measurement of pH was only conducted for bubbling fluid because the volume of porewater is often not sufficient. In addition, the pH of bubbling fluid could not be directly extrapolated to that of subsurface porewater because microbial metabolisms along a redox transition could divert pH into a range distinct from its starting value. For example, anaerobic methane oxidation, the key metabolism in mud volcanoes, converts methane into carbon dioxide that could lead to the enhanced pH. Under the context that the in situ pH could be constrained only with the real measurement, we did not incorporate pH into CCA.

**Comment 5:** Lines 115: How to distinguish between stochastic and deterministic.

**Response:** Please see the response to general comment 1, in which we have performed the calculation of NST and pNST in addition to the existing DDR, NMDS and CCA. The combination of the methods described above provides qualitative and quantitative assessments on the mechanisms controlling biogeographic pattern.

**Comment 6:** Lines 160: There is no information about bacteria orders in Fig.2b.

**Response:** Thanks for the correction. We have removed the description in Fig. 2b.

**Comment 7:** Lines 170: How did the 136 samples come from? The x-axis in Fig.3 seems to be only 15.

**Response:** We did not plot the occurrence of Proteobacteria in all 136 samples in Fig. 3 because the sample size is too big to be properly shown in the figure. Instead, the occurrence was condensed into the site level for the simplicity of presentation. Detailed data regarding the occurrences of individual ASVs (so called ASV table) would be available upon request.

**Comment 8:** Lines 200: The author calculated shannon, chao1 and richness earlier, but why only shannon is studied here.

**Response:** The three diversity indices represent different estimates of community richness. Our calculations yielded that their variation patterns were essentially comparable to each other. Therefore, we only chose the Shannon index for further presentation.

**Comment 9:** Lines 210: Why is geographic distance not included in the CCA analysis. I would suggest authors to consider both physicochemical and geographical factors to reveal the

contribution of environmental and spatial factors to community variation. Based on this, you may state stochastic and deterministic processes in this paper.

**Response:** CCA is used to elucidate the relationships between biological assemblages and environmental characteristics intrinsically attributed to the corresponding niches, thereby facilitating to identify environmental variables that are vital in determining community compositions. In essence, the data required for CCA are all inherited from the attributes of individual samples, such as abundances of specific taxonomic units (e.g., ASV table) or physio-chemical metadata (e.g., temperature, pH). Geographic distance represents a distance gauge between samples, not an intrinsic attribute associated with individual samples. Therefore, it could not be considered as an environmental variable for CCA. Instead, it could be used to constrain the effect of dispersal on community (dis)similarity by relating beta diversity with distance, which is what we did in the original manuscript (shown in Fig. 5). To provide a more quantitative assessment on the stochastic control, we have additionally calculated the NST and pNST for three sample categories, including "bubbling fluid", "surface sediment", and "within site sediment". Please see response 1 for more details.

**Comment 10:** Lines 220: Which graph has a slope of 0.210, I can't find it. Fig.5b: One missing point in the formula.

**Response:** Thanks for the correction. The slope and the figure have been revised.

**Reviewer 2**

This is very interesting and well prepared manuscript. The authors investigated the diversity and composition of microbial communities in the terrestrial mud volcanoes, this is very helpful to enrich our knowledge of the region. However, in terms of analysis methods and expressions in some results, I think there are still shortcomings. The specific opinions are as follows:

**Comment 1:** The stochastic process mentioned in the title is one of the two important processes in community assembly. Why is it only proved by DDR and HAR? Is there any other evidence, such as the use of βNTI and NST (Normalized stochasticity ratio).

**Response:** Thanks for the suggestion. We have calculated the NST index by categorizing all samples into "bubbling fluid", "surface sediment", and "within site sediment". These sample categories represent source communities (for "bubbling fluid"), source communities subject to minor surface impact (for "surface sediment"), and communities potentially altered by localized geochemical context (for "within site sediment"). The results demonstrate a high stochastic control on "bubbling fluid" (NST = 100%) and "surface sediment" (NST = 100%) and a relatively strong control of deterministic process on "within site sediment" (12 out of 15 MVs with NSTs < 50% and the remaining 3 with NSTs between 59% and 63%), a data pattern consistent with the results obtained from existing DDR, NMDS, and CCA analyses. Similar approach has been also applied to the calculation of pNST, which is a measure of stochasticity based on phylogenetic distance between communities and represents an analogy of βNTI. The results demonstrate a pattern similar to that from NST. High stochastic control has been observed for "bubbling fluid" (pNST = 65%) and "surface sediment" (pNST = 73%), whereas mild deterministic control has been inferred for "within site sediment" (8 out of 15 MVs with pNSTs < 50% and the remaining 7 with pNSTs between 51% and 92%). Compared with ASV abundance dissimilarity, phylogenetic dissimilarity leads to a pattern with the stochastic control (pNST > 50%) on more MVs. This could be exemplified with MVs SI and SH where relatively small numbers of ASVs were recovered. The pNSTs for these two MVs are much higher than the corresponding NSTs (44% – 51% versus ~ 1%), suggesting that contrast proportions of phylogenetically distant ASVs between communities lead to a higher degree of random dissimilarity. While our existing analyses are primarily based on the ASV abundance dissimilarity (e.g., NMDS and DDR), we will present both the NSTs and pNSTs results.

**Comment 2:** The article has sequenced the prokaryotic communities, including bacteria and archaea, but why didn't the bacteria and archaea communities be analyzed separately during the analysis? Because the physiological and biochemical characteristics of the two are very different.

**Response:** Thanks for the comment. The proportions of bacteria and archaea varied considerably among samples and sites. In some cases, the prokaryotic communities were almost composed of bacteria. If bacterial and archaeal communities are analyzed separately, the community pattern could have been distorted. Furthermore, some groups of bacteria and archaea interact in different ways under specific geochemical context (e.g., symbiotic partnerships between ANME and SRB at the transition of sulfate to methane). Analyses by merging bacteria and archaeal communities could provide a better assessment on the co-occurrence of different groups and their distribution pattern with geochemical context. Therefore, we did not analyze bacteria and archaea separately.

**Comment 3:** Line 25: This sentence has no real meaning, please omit or revised.

**Response:** Thanks for the suggestion. We have revised the sentence to specifically state that chloride is the most influential parameter for community composition.

**Comment 4:** Line 74: I may have missed this, but where the specific colonists you defined are mentioned?

**Response:** We identified ASVs prevalently distributed or confined within a single MV (Table S2). The most widely distributed ASVs were present in 9 MVs (Fig. 3f; this figure will be modified to magnify the low percentage of prevalently distributed ASVs.) and affiliated with genus of Desulfuromonadaceae or Desulfotignum. In contrast, ~88% of ASVs (28,928 in total) were distributed only in one MV (Fig. 3f). To reduce the complexity of data presentation, we focused on the discussion of these ASVs related to methane and sulfur metabolisms. In particular, the distribution of most abundant or widespread ASVs affiliated with ANME, Desulfobacterales, Methylococcales, and Thiobacillus were presented (Fig. S11). Nevertheless, we will add a supplementary table that comprises the proportion, taxonomic, and site information for 2 most widespread ASVs (across 9 MVs) and 10 most abundant ASVs restricted in any individual MVs.

**Comment 5:** Line 96-99: Please add the specific parameters used in the sequence data analysis.

**Response:** The specific parameters were described in the Supplementary Information.

**Comment 6:** Line 103: As far as I know, only dada2 and deblur softwares can achieve 100% clustering and get the ASV, but you use mothur, how is it achieved?

**Response:** Thanks for the comment. We actually used DADA2 to denoise and identify ASVs first. The representative sequences of all ASVs were subsequently aligned and classified by Mothur. Detail analysis procedures were described in the Supplementary Information.

**Comment 7:** Line 118: What is the parameter $\alpha$ meaning?

**Response:** Thanks for the reminder. "a" is the intercept for the regression. We have revised the description.

**Comment 8:** Line 177: I cannot draw the conclusion from Figure 3f that the most widely distributed ASV were presented in 9 MVs. In addition, the information expressed in the Figure 3 is not fully explained in the text, please revised.

**Response:** Thanks for the comments and correction. We identified a total of 28,928 ASVs across 16 cores distributed in 15 MVs. Only 2 of them were the most widespread and distributed in 9 MVs, whereas other ASVs were recovered from 1 to 6 MVs. The 2 ASVs comprised a fraction of $6.9 \times 10^{-5}$ of the total ASVs. We have revised the figure (Fig. 3f) to magnify the proportions of ASVs distributed in 5 to 9 MVs. We have revised the description about the proportions of different taxonomic units.

**Comment 9:** Line 219: The result of Mantel test is negative, is this correct?

**Response:** Thanks for the correction. The value should be positive. We have revised the value.